# PyNeRF: Pyramidal Neural Radiance Fields

**Haithem Turki**
Carnegie Mellon University
hturki@cs.cmu.edu

**Michael Zollhöfer**
Meta Reality Labs Research
zollhoefer@meta.com

**Christian Richardt**
Meta Reality Labs Research
crichardt@meta.com

**Deva Ramanan**
Carnegie Mellon University
deva@cs.cmu.edu

## Abstract

Neural Radiance Fields (NeRFs) can be dramatically accelerated by spatial grid representations [6, 9, 20, 25]. However, they do not explicitly reason about scale and so introduce aliasing artifacts when reconstructing scenes captured at different camera distances. Mip-NeRF and its extensions propose scale-aware renderers that project volumetric frustums rather than point samples but such approaches rely on positional encodings that are not readily compatible with grid methods. We propose a simple modification to grid-based models by training model heads at different spatial grid resolutions. At render time, we simply use coarser grids to render samples that cover larger volumes. Our method can be easily applied to existing accelerated NeRF methods and significantly improves rendering quality (reducing error rates by 20–90% across synthetic and unbounded real-world scenes) while incurring minimal performance overhead (as each model head is quick to evaluate). Compared to Mip-NeRF, we reduce error rates by 20% while training over 60× faster.

## 1   Introduction

Recent advances in neural volumetric rendering techniques, most notably around Neural Radiance Fields [19] (NeRFs), have lead to significant progress towards photo-realistic novel view synthesis. However, although NeRF provides state-of-the-art rendering quality, it is notoriously slow to train and render due in part to its internal MLP representation. It further assumes that scene content is equidistant from the camera and rendering quality degrades due to aliasing and excessive blurring when that assumption is violated.

Recent methods [6, 9, 20, 25] accelerate NeRF training and rendering significantly through the use of grid representations. Others, such as Mip-NeRF [2], address aliasing by projecting camera frustum volumes instead of point-sampling rays. However, these anti-aliasing methods rely on the base NeRF MLP representation (and are thus slow) and are incompatible with grid representations due to their reliance on non-grid-based inputs.

Inspired by divide-and-conquer NeRF extensions [22, 23, 27, 30] and classical approaches such as Gaussian pyramids [1] and mipmaps [34], we propose a simple approach that can easily be applied to any existing accelerated NeRF implementation. We train a pyramid of models at different scales, sample along camera rays (as in the original NeRF), and simply query coarser levels of the pyramid for samples that cover larger volumes (similar to voxel cone tracing [8]). Our method is simple to implement and significantly improves the rendering quality of fast rendering approaches with minimal performance overhead.

37th Conference on Neural Information Processing Systems (NeurIPS 2023).

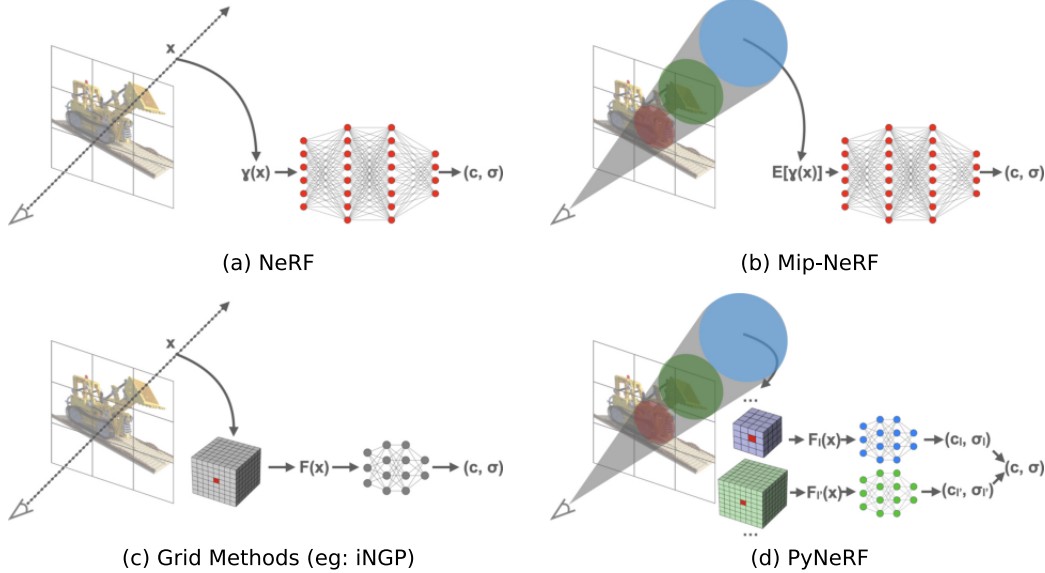

(a) NeRF

(b) Mip-NeRF

(c) Grid Methods (eg: iNGP)

(d) PyNeRF

Figure 1: **Comparison of methods. (a)** NeRF traces a ray from the camera's center of projection through each pixel and samples points $\mathbf{x}$ along each ray. Sample locations are then encoded with a positional encoding to produce a feature $\gamma(\mathbf{x})$ that is fed into an MLP. **(b)** Mip-NeRF instead reasons about *volumes* by defining a 3D conical frustum per camera pixel. It splits the frustum into sampled volumes, approximates them as multivariate Gaussians, and computes the integral of the positional encodings of the coordinates contained within the Gaussians. Similar to NeRF, these features are then fed into an MLP. **(c)** Accelerated grid methods, such as iNGP, sample points as in NeRF, but do not use positional encoding and instead featurize each point by interpolating between vertices in a feature grid. These features are then passed into a much smaller MLP, which greatly accelerates training and rendering. **(d)** PyNeRF also uses feature grids, but reasons about volumes by training separate models at different scales (similar to a mipmap). It calculates the area covered by each sample in world coordinates, queries the models at the closest corresponding resolutions, and interpolates their outputs.

**Contribution:** Our primary contribution is a partitioning method that can be easily adapted to any existing grid-rendering approach. We present state-of-the-art reconstruction results against a wide range of datasets, including on novel scenes we designed that explicitly target common aliasing patterns. We evaluate different posssible architectures and demonstrate that our design choices provide a high level of visual fidelity while maintaining the rendering speed of fast NeRF approaches.

## 2 Related Work

The now-seminal Neural Radiance Fields (NeRF) paper [19] inspired a vast corpus of follow-up work. We discuss a non-exhaustive list of such approaches along axes relevant to our work.

**Grid-based methods.** The original NeRF took 1–2 days to train, with extensions for unbounded scenes [3, 40] taking longer. Once trained, rendering takes seconds per frame and is far below interactive thresholds. NSVF [17] combines NeRF's implicit representation with a voxel octree that allows for empty-space skipping and improves inference speeds by 10×. Follow-up works [10, 11, 39] further improve rendering to interactive speeds by storing precomputed model outputs into auxiliary grid structures that bypass the need to query the original model altogether at render time. Plenoxels [25] and DVGO [26] accelerate both training and rendering by directly optimizing a voxel grid instead of an MLP to train in minutes or even seconds. TensoRF [6] and K-Planes [9] instead use the product of low-rank tensors to approximate the voxel grid and reduce memory usage, while Instant-NGP [20] (iNGP) encodes features into a multi-resolution hash table. The main goal of our work is to combine the speed benefits of grid-based methods with an approach that maintains quality across different rendering scales.

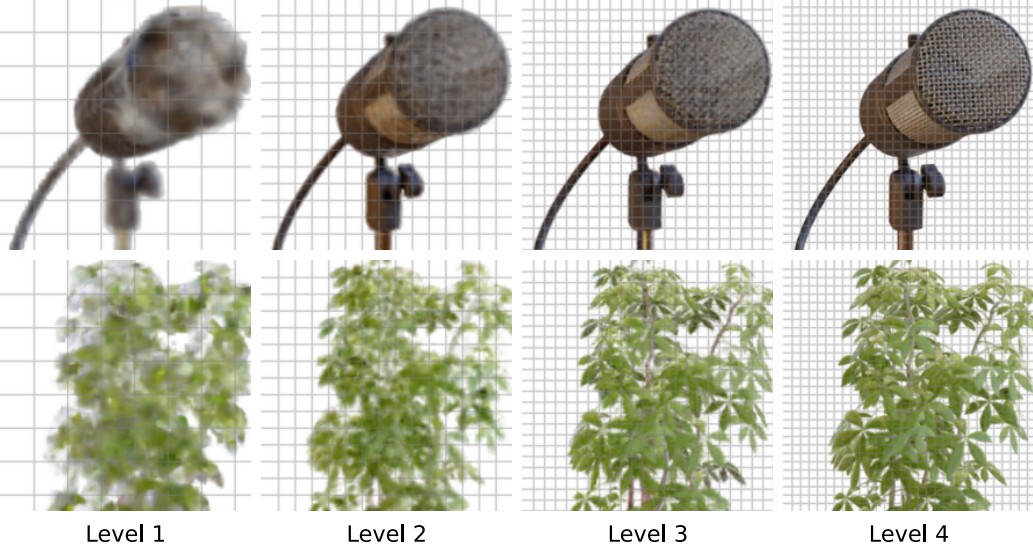

| Level 1 | Level 2 | Level 3 | Level 4 |

Figure 2: We visualize renderings from a pyramid of spatial grid-based NeRFs trained for different voxel resolutions. Models at finer pyramid levels tend to capture finer content.

**Divide-and-conquer.** Several works note the diminishing returns in using large networks to represent scene content, and instead render the area of interest with multiple smaller models. DeRF [22] and KiloNeRF [23] focus on inference speed while Mega-NeRF [30], Block-NeRF [27], and SUDS [31] use scene decomposition to efficiently train city-scale neural representations. Our method is similar in philosophy, although we partition across different resolutions instead of geographical area.

**Aliasing.** The original NeRF assumes that scene content is captured at roughly equidistant camera distances and emits blurry renderings when the assumption is violated. Mip-NeRF [2] reasons about the volume covered by each camera ray and proposes an integrated positional encoding that alleviates aliasing. Mip-NeRF 360 [3] extends the base method to unbounded scenes. Exact-NeRF [14] derives a more precise integration formula that better reconstructs far-away scene content. Bungee-NeRF [36] leverages Mip-NeRF and further adopts a coarse-to-fine training approach with residual blocks to train on large-scale scenes with viewpoint variation. LIRF [37] proposes a multiscale image-based representation that can generalize across scenes. The methods all build upon the original NeRF MLP model and do not readily translate to accelerated grid-based methods.

**Concurrent work.** Several contemporary efforts explore the intersection of anti-aliasing and fast rendering. Zip-NeRF [4] combines a hash table representation with a multi-sampling method that approximates the true integral of features contained within each camera ray's view frustum. Although it trains faster than Mip-NeRF, it is explicitly not designed for fast rendering as the multi-sampling adds significant overhead. Mip-VoG [12] downsamples and blurs a voxel grid according to the volume of each sample in world coordinates. We compare their reported numbers to ours in Section 4.2. Tri-MipRF [13] uses a similar prefiltering approach, but with triplanes instead of a 3D voxel grid.

**Classical methods.** Similar to PyNeRF, classic image processing methods, such as Gaussian [1] and Laplacian [5] hierarchy, maintain a coarse-to-fine pyramid of different images at different resolutions. Compared to Mip-NeRF, which attempts to learn a single MLP model across all scales, one could argue that our work demonstrates that the classic pyramid approach can be efficiently adapted to neural volumetric models. In addition, our ray sampling method is similar to Crassin et al.'s approach [8], which approximates cone tracing by sampling along camera rays and querying different mipmap levels according the spatial footprint of each sample (stored as a voxel octree in their approach and as a NeRF model in ours).

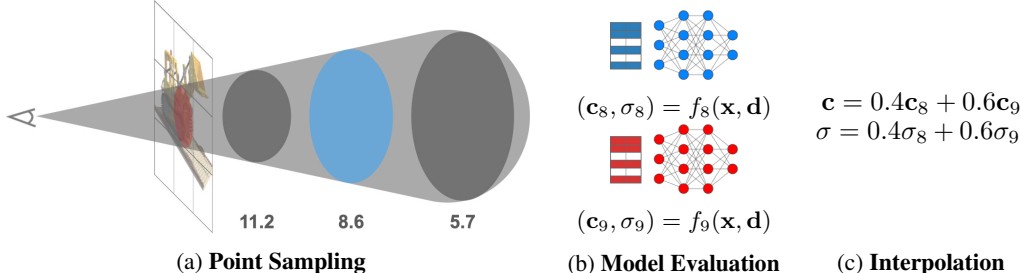

$(\mathbf{c}_8, \sigma_8) = f_8(\mathbf{x}, \mathbf{d})$

$\mathbf{c} = 0.4\mathbf{c}_8 + 0.6\mathbf{c}_9$
$\sigma = 0.4\sigma_8 + 0.6\sigma_9$

$(\mathbf{c}_9, \sigma_9) = f_9(\mathbf{x}, \mathbf{d})$

(a) **Point Sampling**      (b) **Model Evaluation**      (c) **Interpolation**

Figure 3: **Overview.** (**a**) We sample frustums along the camera ray corresponding to each pixel and derive the scale of each sample according to its width in world coordinates. (**b**) We query the model heads closest to the scale of each sample. (**c**) We derive a single color and density value for each sample by interpolating between model outputs according to scale.

## 3 Approach

### 3.1 Preliminaries

**NeRF.** NeRF [19] represents a scene within a continuous volumetric radiance field that captures geometry and view-dependent appearance. It encodes the scene within the weights of a multi-layer perceptron (MLP). At render time, NeRF casts a camera ray $\mathbf{r}$ for each image pixel. NeRF samples multiple positions $\mathbf{x}_i$ along each ray and queries the MLP at each position (along with the ray viewing direction $\mathbf{d}$) to obtain density and color values $\sigma_i$ and $\mathbf{c}_i$. To better capture high-frequency details, NeRF maps $\mathbf{x}_i$ and $\mathbf{d}$ through an $L$-dimensional positional encoding (PE) $\gamma(x) = [\sin(2^0\pi x), \cos(2^0\pi x), \ldots, \sin(2^L\pi x), \cos(2^L\pi x)]$ instead of directly using them as MLP inputs. It then composites a single color prediction $\hat{C}(\mathbf{r})$ per ray using numerical quadrature $\sum_{i=0}^{N-1} T_i(1 - \exp(-\sigma_i\delta_i))\, c_i$, where $T_i = \exp(-\sum_{j=0}^{i-1} \sigma_j\delta_j)$ and $\delta_i$ is the distance between samples. The training process optimizes the model by sampling batches $\mathcal{R}$ of image pixels and minimizing the loss $\sum_{\mathbf{r} \in \mathcal{R}} \left\| C(\mathbf{r}) - \hat{C}(\mathbf{r}) \right\|^2$. We refer the reader to Mildenhall et al. [19] for details.

**Anti-aliasing.** The original NeRF suffers from aliasing artifacts when reconstructing scene content observed at different distances or resolutions due to its reliance on point-sampled features. As these features ignore the volume viewed by each ray, different cameras viewing the same position from different distances may produce the same ambiguous feature. Mip-NeRF [2] and variants instead reason about *volumes* by defining a 3D conical frustum per camera pixel. It featurizes intervals within the frustum with a integrated positional encoding (IPE) that approximates each frustum as a multivariate Gaussian to estimate the integral $\mathbb{E}[\gamma(x)]$ over the PEs of the coordinates within it.

**Grid-based acceleration.** Various methods [6, 9, 20, 25, 26] eschew NeRF's positional encoding and instead store learned features into a grid-based structure, e.g. implemented as an explicit voxel grid, hash table, or a collection of low-rank tensors. The features are interpolated based on the position of each sample and then passed into a hard-coded function or much smaller MLP to produce density and color, thereby accelerating training and rendering by orders of magnitude. However, these approaches all use the same volume-insensitive point sampling of the original NeRF and do not have a straightforward analogy to Mip-NeRF's IPE as they no longer use positional encoding.

### 3.2 Multiscale sampling

Assume that each sample $\mathbf{x}$ (where we drop the $i$ index to reduce notational clutter) is associated with an integration volume. Intuitively, samples close to a camera correspond to small volumes, while samples far away from a camera correspond to large volumes (Figure 3). Our crucial insight for enabling multiscale sampling with grid-based approaches is remarkably simple: *we train separate NeRFs at different voxel resolutions and simply use coarser NeRFs for samples covering larger volumes.* Specifically, we define a hierarchy of $L$ resolutions that divide the world into voxels of length $1/N_0, ..., 1/N_{L-1}$, where $N_{l+1} = sN_l$ and $s$ is a constant scaling factor. We also define a function $f_l(\mathbf{x}, \mathbf{d})$ at each level that maps from sample location $\mathbf{x}$ and viewing direction $\mathbf{d}$ to color $\mathbf{c}$ and density $\sigma$. $f_l$ can be implemented by any grid-based NeRF; in our experiments, we use a

---

**Algorithm 1** PyNeRF rendering function

---

**Input:** $m$ rays $\mathbf{r}$, $L$ pyramid levels, hierarchy mapping function $M$, base resolution $N_0$, scaling factor $s$

**Output:** $m$ estimated colors $\mathbf{c}$

    $\mathbf{x}, \mathbf{d}, P(\mathbf{x}) \leftarrow sample(\mathbf{r})$         ▷ Sample points $\mathbf{x}$ along each ray with direction $\mathbf{d}$ and area $P(\mathbf{x})$

    $M(P(\mathbf{x})) \leftarrow \log_s(P(\mathbf{x})/N_0)$         ▷ Equation 1

    $l \leftarrow \min(L-1, \max(0, \lceil M(P(\mathbf{x})) \rceil))$         ▷ Equation 2

    $w \leftarrow l - M(P(\mathbf{x}))$         ▷ Equation 5

    $model\_out \leftarrow zeros(len(\mathbf{x}))$         ▷ Zero-initialize model outputs for each sample $\mathbf{x}$

    **for** $i$ in unique($l$) **do**         ▷ Iterate over sample levels

        $model\_out[l=i]\ += w[l=i]f_i(\mathbf{x}[l=i], \mathbf{d}[l=i])$

        $model\_out[l=i]\ += (1-w)[l=i]f_{i-1}(\mathbf{x}[l=i], \mathbf{d}[l=i])$

    **end for**

    $\mathbf{c} \leftarrow composite(model\_out)$         ▷ Composite model outputs into per-ray color $\mathbf{c}$

    **return c**

---

hash table followed by small density and color MLPs, similar to iNGP. We further define a mapping function $M$ that assigns the integration volume of sample $\mathbf{x}$ to the hierarchy level $l$. We explore different alternatives, but find that selecting the level whose voxels project to the 2D pixel area $P(\mathbf{x})$ used to define the integration volume works well:

$$M(P(\mathbf{x})) = \log_s(P(\mathbf{x})/N_0) \tag{1}$$

$$l = \min(L-1, \max(0, \lceil M(P(\mathbf{x})) \rceil)) \tag{2}$$

$$\sigma, \mathbf{c} = f_l(\mathbf{x}, \mathbf{d}), \qquad \textbf{[GaussPyNeRF]} \tag{3}$$

where $\lceil \cdot \rceil$ is the ceiling function. Such a model can be seen as a (Gaussian) pyramid of spatial grid-based NeRFs (Fig. 2). If the final density and color were obtained by *summing* across different pyramid levels, the resulting levels would learn to specialize to residual or "band-pass" frequencies (as in a 3D Laplacian pyramid [5]):

$$\sigma, \mathbf{c} = \sum_{i=0}^{l} f_i(\mathbf{x}, \mathbf{d}). \qquad \textbf{[LaplacianPyNeRF]} \tag{4}$$

Our experiments show that such a representation is performant, but expensive since it requires $l$ model evaluations per sample. Instead, we find a good tradeoff is to linearly interpolate between two model evaluations at the levels just larger than and smaller than the target integration volume:

$$\sigma, \mathbf{c} = wf_l(\mathbf{x}, \mathbf{d}) + (1-w)f_{l-1}(\mathbf{x}, \mathbf{d}), \quad \text{where} \quad w = l - M(P(\mathbf{x})). \quad \textbf{(Default) [PyNeRF]} \tag{5}$$

This adds the cost of only a *single* additional evaluation (increasing the overall rendering time from 0.0045 to 0.005 ms per pixel) while maintaining rendering quality (see Section 4.6). Our algorithm is summarized in Algorithm 1.

**Matching areas vs volumes.** One might suspect it may be better to select the voxel level $l$ whose volume best matches the sample's 3D integration volume. We experimented with this, but found it more effective to match the projected 2D pixel area rather than volumes. Note that both approaches would produce identical results if the 3D volume was always a cube, but volumes may be elongated along the ray depending on the sampling pattern. Matching areas is preferable because most visible 3D scenes consist of empty space and surfaces, implying that when computing the composite color for a ray $r$, most of the contribution will come from a few samples $\mathbf{x}$ lying near the surface of intersection. When considering the target 3D integration volume associated with $\mathbf{x}$, most of the contribution to the final composite color will come from integrating along the 2D surface (since the rest of the 3D volume is either empty or hidden). This loosely suggests we should select levels of the voxel hierarchy based on (projected) area rather than volume.

**Hierarchical grid structures.** Our method can be applied to any accelerated grid method irrespective of the underyling storage. However, a drawback of this approach is an increased on-disk serialization footprint due to training a hierarchy of spatial grid NeRFs. A possible solution is to exploit hierarchical grid structures that already exist *within* the base NeRF. Note that multi-resolution grids such as those

Table 1: **Synthetic results.** PyNeRF outperforms all baselines and trains over 60× faster than Mip-NeRF. Both PyNeRF and Mip-NeRF properly reconstruct the brick wall in the Blender-A dataset, but Mip-NeRF fails to accurately reconstruct checkerboard patterns.

| | Multiscale Blender [2] | | | | | Blender-A | | | | |
|---|---|---|---|---|---|---|---|---|---|---|
| | ↑PSNR | ↑SSIM | ↓LPIPS | ↓Avg Error | ↓Train Time (h) | ↑PSNR | ↑SSIM | ↓LPIPS | ↓Avg Error | ↓ Train Time (h) |
| Plenoxels [25] | 24.98 | 0.843 | 0.161 | 0.080 | 0:28 | 18.13 | 0.511 | 0.523 | 0.190 | **0:40** |
| K-Planes [9] | 29.88 | 0.946 | 0.058 | 0.022 | 0:32 | 21.17 | 0.593 | 0.641 | 0.405 | 1:22 |
| TensoRF [6] | 30.04 | 0.948 | 0.056 | 0.021 | 0:27 | 27.01 | 0.785 | 0.197 | 0.054 | 1:20 |
| iNGP [20] | 30.21 | 0.958 | 0.040 | 0.022 | **0:20** | 20.85 | 0.767 | 0.244 | 0.089 | 0:56 |
| Nerfacto [28] | 29.56 | 0.947 | 0.051 | 0.022 | 0:25 | 27.46 | 0.796 | 0.195 | 0.053 | 1:07 |
| Mip-VoG [12] | 30.42 | 0.954 | 0.053 | — | — | — | — | — | — | — |
| Mip-NeRF [2] | 34.50 | 0.974 | 0.017 | 0.009 | 29:49 | 31.33 | 0.894 | 0.098 | 0.063 | 30:12 |
| PyNeRF | **34.78** | **0.976** | **0.015** | **0.008** | 0:25 | **41.99** | **0.986** | **0.007** | **0.004** | 1:10 |

used by iNGP [20] or K-Planes [9] already define a scale hierarchy that is a natural fit for PyNeRF. Rather than learning a separate feature grid for each model in our pyramid, we can reuse the same multi-resolution features across levels (while still training different MLP heads).

**Multi-resolution pixel input.** One added benefit of the above is that one can train with multiscale training data, which is particularly helpful for learning large, city-scale NeRFs [27, 30, 31, 36, 38]. For such scenarios, even storing high-resolution pixel imagery may be cumbersome. In our formulation, one can store low-resolution images and quickly train a coarse scene representation. The benefits are multiple. Firstly, divide-and-conquer approaches such as Mega-NeRF [31] partition large scenes into smaller cells and train using different training pixel/ray subsets for each (to avoid training on irrelevant data). However, in the absence of depth sensors or a priori 3D scene knowledge, Mega-NeRF is limited in its ability to prune irrelevant pixels/rays (due to intervening occluders) which empirically bloat the size of each training partition by 2× [30]. With our approach, we can learn a coarse 3D knowledge of the scene on downsampled images and then filter higher-resolution data partitions more efficiently. Once trained, lower-resolution levels can also serve as an efficient initialization for finer layers. In addition, many contemporary NeRF methods use occupancy grids [20] or proposal networks [3] to generate refined samples near surfaces. We can quickly train these along with our initial low-resolution model and then use them to train higher-resolution levels in a sample-efficient manner. We show in our experiments that such course-to-fine multiscale training can speed up convergence (Section 4.5).

**Unsupervised levels.** A naive implementation of our method will degrade when zooming in and out of areas that have not been seen at training time. Our implementation mitigates this by maintaining an auxiliary data structure (similar to an occupancy grid [20]) that tracks the coarsest and finest levels queried in each region during training. We then use the structure at inference time to only query levels that were supervised during training.

## 4 Experiments

We first evaluate PyNeRF's performance by measuring its reconstruction quality on bounded synthetic (Section 4.2) and unbounded real-world (Section 4.3) scenes. We demonstrate PyNeRF's generalizability by evaluating it on additional NeRF backbones (Section 4.4) and then explore the convergence benefits of using multiscale training data in city-scale reconstruction scenarios (Section 4.5). We ablate our design decisions in Section 4.6.

### 4.1 Experimental Setup

**Training.** We implement PyNeRF on top of the Nerfstudio library [28] and train on each scene with 8,192 rays per batch by default for 20,000 iterations on the Multiscale Blender and Mip-NeRF 360 datasets, and 50,000 iterations on the Boat dataset and Blender-A. We train a hierarchy of 8 PyNeRF levels backed by a single multi-resolution hash table similar to that used by iNGP [20] in Section 4.2 and Section 4.3 before evaluating additional backbones in Section 4.4. We use 4 features per level with a hash table size of $2^{20}$ by default, which we found to give the best quality-performance trade-off on the A100 GPUs we use in our experiments. Each PyNeRF uses a 64-channel density MLP with one hidden layer followed by a 128-channel color MLP with two hidden layers. We use similar model capacities in our baselines for fairness. We sample rays using an occupancy

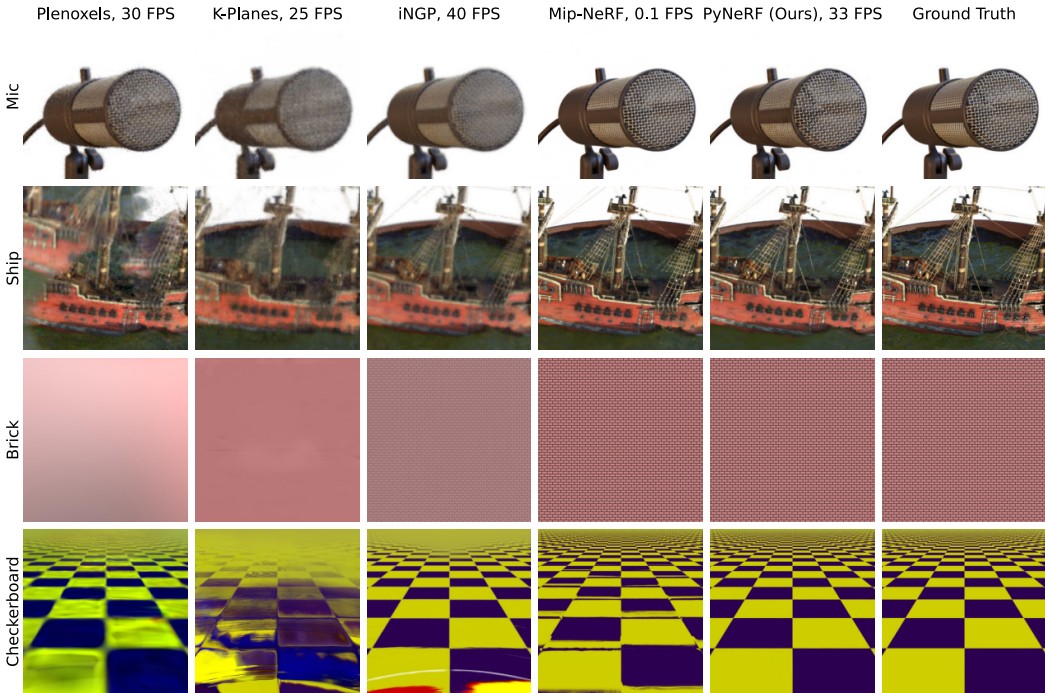

Figure 4: **Synthetic results.** PyNeRF and Mip-NeRF provide comparable results on the first three scenes that are crisper than those of the other fast renderers. Mip-NeRF does not accurately render the tiles in the last row while PyNeRF recreates them near-perfectly.

grid [20] on the Multiscale Blender dataset, and with a proposal network [3] on all others. We use gradient scaling [21] to improve training stability in scenes with that capture content at close distance (Blender-A and Boat). We parameterize unbounded scenes with Mip-NeRF 360's contraction method.

**Metrics.** We report quantitative results based on PSNR, SSIM [33], and the AlexNet implementation of LPIPS [41], along with the training time in hours as measured on a single A100 GPU. For ease of comparison, we also report the "average" error metric proposed by Mip-NeRF [2] composed of the geometric mean of $\mathrm{MSE} = 10^{-\mathrm{PSNR}/10}$, $\sqrt{1 - \mathrm{SSIM}}$, and LPIPS.

## 4.2 Synthetic Reconstruction

**Datasets.** We evaluate PyNeRF on the Multiscale Blender dataset proposed by Mip-NeRF along with our own Blender scenes (which we name "Blender-A") intended to further probe the anti-aliasing ability of our approach (by reconstructing a slanted checkerboard and zooming into a brick wall).

**Baselines.** We compare PyNeRF to several fast-rendering approaches, namely Instant-NGP [20] and Nerfacto [28], which store features within a multi-resolution hash table, Plenoxels [25] which optimizes an explicit voxel grid, and TensoRF [6] and K-Planes [9], which rely on low-rank tensor decomposition. We also compare our Multiscale Blender results to those reported by Mip-VoG [12], a contemporary fast anti-aliasing approach, and to Mip-NeRF [2] on both datasets.

**Results.** We summarize our results in Table 1 and show qualitative examples in Figure 4. PyNeRF outperforms all fast rendering approaches as well as Mip-VoG by a wide margin and is slightly better than Mip-NeRF on Multiscale Blender while training over 60× faster. Both PyNeRF and Mip-NeRF properly reconstruct the brick wall in the Blender-A dataset, but Mip-NeRF fails to accurately reconstruct checkerboard patterns.

## 4.3 Real-World Reconstruction

**Datasets.** We evaluate PyNeRF on the Boat scene of the ADOP [24] dataset, which to our knowledge is one of the only publicly available unbounded real-world captures that captures its primary object of

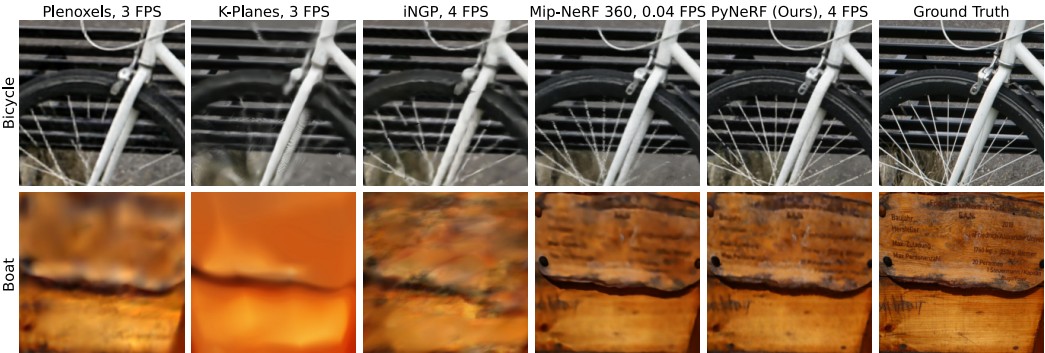

Figure 5: **Real-world results.** PyNeRF reconstructs higher-fidelity details (such as the spokes on the bicycle and the lettering within the boat) than other methods.

Table 2: **Real-world results.** PyNeRF outperforms all baselines in PSNR and average error, and trains 40× faster than Mip-NeRF 360 and 100× faster than Exact-NeRF (the next best methods).

| | Boat [24] | | | | | Mip-NeRF 360 [3] | | | | |
|---|---|---|---|---|---|---|---|---|---|---|
| | ↑PSNR | ↑SSIM | ↓LPIPS | ↓Avg Error | ↓Train Time (h) | ↑PSNR | ↑SSIM | ↓LPIPS | ↓Avg Error | ↓ Train Time (h) |
| Plenoxels [25] | 17.05 | 0.505 | 0.617 | 0.185 | 2:14 | 21.88 | 0.606 | 0.524 | 0.117 | 1:00 |
| K-Planes [9] | 18.00 | 0.501 | 0.590 | 0.168 | 2:41 | 21.53 | 0.577 | 0.500 | 0.120 | 1:08 |
| TensoRF [6] | 14.75 | 0.398 | 0.630 | 0.234 | 2:30 | 18.07 | 0.439 | 0.677 | 0.181 | 1:07 |
| iNGP [20] | 15.34 | 0.433 | 0.646 | 0.222 | **1:42** | 21.14 | 0.568 | 0.521 | 0.126 | **0:40** |
| Nerfacto [28] | 19.27 | 0.570 | 0.425 | 0.135 | 2:12 | 22.47 | 0.616 | 0.431 | 0.105 | 1:02 |
| Mip-NeRF 360 w/ GLO [3] | 20.03 | 0.595 | **0.416** | 0.124 | 37:28 | 22.76 | **0.664** | **0.342** | **0.095** | 37:35 |
| Mip-NeRF 360 w/o GLO [3] | 15.92 | 0.480 | 0.501 | 0.194 | 37:10 | 22.70 | **0.664** | **0.342** | 0.095 | 37:22 |
| Exact-NeRF w/ GLO [14] | 20.21 | **0.601** | 0.425 | 0.123 | 109:11 | 21.40 | 0.619 | 0.416 | 0.121 | 110:06 |
| Exact-NeRF w/o GLO [14] | 16.33 | 0.489 | 0.510 | 0.187 | 107:52 | 22.56 | 0.619 | 0.410 | 0.121 | 108:11 |
| PyNeRF | **20.43** | **0.601** | 0.422 | **0.121** | 2:12 | **23.09** | 0.654 | 0.358 | **0.094** | 1:00 |

interest from different camera distances. For further comparison, we construct a multiscale version of the outdoor scenes in the Mip-NeRF 360 [3] dataset using the same protocol as Multiscale Blender [2].

**Baselines.** We compare PyNeRF to the same fast-rendering approaches as in Section 4.2, along with two unbounded Mip-NeRF variants: Mip-NeRF 360 [3] and Exact-NeRF [14]. We report numbers on each variant with and without generative latent optimization [18] to account for lighting changes.

**Results.** We summarize our results in Table 2 along with qualitative results in Figure 5. Once again, PyNeRF outperforms all baselines, trains 40× faster than Mip-NeRF 360, and 100× faster than Exact-NeRF (the next best alternatives).

### 4.4  Additional Backbones

**Methods.** We demonstrate how PyNeRF can be applied to any grid-based NeRF method by evaluating it with K-Planes [9] and TensoRF [6] in addition to our default iNGP-based implementatino. We take advantage of the inherent multi-resolution structure of iNGP and K-Planes by reusing the same feature grid across PyNeRF levels and train a separate feature grid per level in our TensoRF variant.

**Results.** We train the PyNeRF variants along with their backbones across the datasets described in Section 4.2 and Section 4.3, and summarize the results in Table 3. All PyNeRF variants show clear improvements over their base methods.

### 4.5  City-Scale Convergence

**Dataset.** We evaluate PyNeRF's convergence properties on the the Argoverse 2 [35] Sensor dataset (to our knowledge, the largest city-scale dataset publicly available). We select the largest overlapping subset of logs and filter out moving objects through a pretrained segmentation model [7]. The resulting training set contains 400 billion rays across 150K video frames.

**Methods.** We use SUDS [31] as the backbone model in our experiments. We begin training our method on 8× downsampled images (containing 64× fewer rays) for 5,000 iterations and then on

Table 3: **Additional backbones.** We train the PyNeRF variants along with their backbones across the datasets described in Section 4.2 and Section 4.3 All PyNeRF variants outperform their baselines by a wide margin.

| | Synthetic | | | | Real-World | | | |
|---|---|---|---|---|---|---|---|---|
| | ↑PSNR | ↑SSIM | ↓LPIPS | ↓Avg Error | ↑PSNR | ↑SSIM | ↓LPIPS | ↓Avg Error |
| iNGP [20] | 28.86 | 0.916 | 0.087 | 0.032 | 19.94 | 0.541 | 0.537 | 0.146 |
| K-Planes [9] | 27.90 | 0.865 | 0.131 | 0.047 | 20.54 | 0.553 | 0.520 | 0.136 |
| TensoRF [6] | 29.12 | 0.902 | 0.100 | 0.042 | 17.21 | 0.421 | 0.696 | 0.200 |
| PyNeRF | **36.22** | **0.979** | **0.013** | **0.004** | **22.65** | **0.645** | **0.369** | **0.098** |
| PyNeRF-K-Planes | 35.42 | 0.975 | 0.014 | 0.005 | 22.00 | 0.622 | 0.405 | 0.108 |
| PyNeRF-TensoRF | 35.67 | 0.976 | 0.015 | 0.005 | 21.35 | 0.568 | 0.482 | 0.122 |

Table 4: **City-scale convergence.** We track rendering quality over the first four hours of training. PyNeRF achieves the same rendering quality as SUDS 2× faster.

| | ↑ PSNR | | | | | ↑ SSIM | | | |
|---|---|---|---|---|---|---|---|---|---|
| Time (h) | 1:00 | 2:00 | 3:00 | 4:00 | Time (h) | 1:00 | 2:00 | 3:00 | 4:00 |
| SUDS [31] | 16.01 | 17.41 | 18.08 | 18.53 | SUDS [31] | 0.570 | 0.600 | 0.602 | 0.606 |
| PyNeRF | **17.17** | **18.44** | **18.59** | **18.73** | PyNeRF | **0.614** | **0.618** | **0.619** | **0.621** |

| | ↓ LPIPS | | | | | ↓ Avg Error | | | |
|---|---|---|---|---|---|---|---|---|---|
| Time (h) | 1:00 | 2:00 | 3:00 | 4:00 | Time (h) | 1:00 | 2:00 | 3:00 | 4:00 |
| SUDS [31] | 0.531 | 0.496 | 0.470 | 0.466 | SUDS [31] | 0.182 | 0.160 | 0.150 | 0.145 |
| PyNeRF | **0.521** | **0.485** | **0.469** | **0.465** | PyNeRF | **0.165** | **0.146** | **0.144** | **0.142** |

progressively higher resolutions (downsampled to 4×, 2×, and 1×) every 5,000 iterations hereafter. We compare to the original SUDS method as a baseline.

**Metrics.** We report the evolution of the quality metrics used in Section 4.2 and Section 4.3 over the first four hours of the training process.

**Results.** We summarize our results in Table 4. PyNeRF converges more rapidly than the SUDS baseline, achieving the same rendering quality at 2 hours as SUDS after 4.

## 4.6 Diagnostics

**Methods.** We validate our design decisions by testing several variants. We ablate our MLP-level interpolation described in Equation 5 and compare it to the GausssPyNeRF and LaplacianPyNeRF variants described in Section 3.2 along with another that instead interpolates the learned grid feature vectors (which avoids the need for an additional MLP evaluation per sample). As increased storage footprint is a potential drawback method, we compare our default strategy of sharing the same multi-resolution feature grid across PyNeRF levels to the naive implementation that trains a separate grid per level. We also explore using 3D sample volumes instead of projected 2D pixel areas to determine voxel levels $l$.

**Results.** We train our method and variants as described in Section 4.2 and Section 4.3, and summarize the results (averaged across datasets) in Table 5. Our proposed interpolation method strikes a good balance — its performance is near-identical to the full LaplacianPyNeRF approach while training 3× faster (and is significantly better than the other interpolation methods). Our strategy of reusing the same feature grid across levels performs comparably to the naive implementation while training faster due to fewer feature grid lookups. Using 2D pixel areas instead of 3D volumes to determine voxel level $l$ provides an improvement.

Table 5: **Diagnostics.** The rendering quality of our interpolation method is near-identical to the full residual approach while training 3× faster, and is significantly better than other alternatives. Reusing the same feature grid across levels performs comparably to storing separate hash tables per level while training faster.

| Method | Our Interp. | Shared Features | 2D Area | ↑PSNR | ↑SSIM | ↓LPIPS | ↓ Avg Error | ↓ Train Time (h) |
|---|---|---|---|---|---|---|---|---|
| GaussPyNeRF (Eq. 3) | ✗ | ✓ | ✓ | 28.72 | 0.803 | 0.201 | 0.056 | **0:43** |
| LaplacianPyNeRF (Eq. 4) | ✗ | ✓ | ✓ | **29.48** | **0.813** | **0.190** | **0.052** | 2:44 |
| Feature grid interpolation | ✗ | ✗ | ✓ | 28.45 | 0.767 | 0.244 | 0.070 | 0:46 |
| Separate hash tables | ✓ | ✗ | ✓ | 29.41 | **0.813** | 0.196 | 0.054 | 0:52 |
| Levels w/ 3D Volumes | ✓ | ✓ | ✗ | 29.19 | 0.811 | 0.184 | 0.054 | 0:48 |
| PyNeRF | ✓ | ✓ | ✓ | 29.44 | 0.812 | 0.191 | 0.053 | 0:48 |

## 5 Limitations

Although our method generalizes to any grid-based method (Section 4.4), it requires a larger on-disk serialization footprint due to training a hierarchy of spatial grid NeRFs. This can be mitigated by reusing the same feature grid when the underlying backbone uses a multi-resolution feature grid [9, 20], but this is not true of all methods [6, 25].

## 6 Societal Impact

Our method facilitates the rapid construction of high-quality neural representations in a resource efficient manner. As such, the risks inherent to our work is similar to those of other neural rendering papers, namely privacy and security concerns related to the intentional or inadvertent capture or privacy-sensitive information such as human faces and vehicle license plate numbers. While we did not apply our approach to data with privacy or security concerns, there is a risk, similar to other neural rendering approaches, that such data could end up in the trained model if the employed datasets are not properly filtered before use. Many recent approaches [15, 16, 29, 31, 42] distill semantics into NeRF's representation, which may be used to filter out sensitive information at render time. However this information would still reside in the model itself. This could in turn be mitigated by preprocessing the input data used to train the model [32].

## 7 Conclusion

We propose a method that significantly improves the anti-aliasing properties of fast volumetric renderers. Our approach can be easily applied to any existing grid-based NeRF, and although simple, provides state-of-the-art reconstruction results against a wide variety of datasets (while training 60–100× faster than existing anti-aliasing methods). We propose several synthetic scenes that model common aliasing patterns as few existing NeRF datasets cover these scenarios in practice. Creating and sharing additional real-world captures would likely facilitate further research.

**Acknowledgements.** HT and DR were supported in part by the Intelligence Advanced Research Projects Activity (IARPA) via Department of Interior/ Interior Business Center (DOI/IBC) contract number 140D0423C0074. The U.S. Government is authorized to reproduce and distribute reprints for Governmental purposes notwithstanding any copyright annotation thereon. Disclaimer: The views and conclusions contained herein are those of the authors and should not be interpreted as necessarily representing the official policies or endorsements, either expressed or implied, of IARPA, DOI/IBC, or the U.S. Government.

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
