# Supplemental Materials

## A  Single-scale datasets

Although PyNeRF is designed for scenarios that capture scene content at different distances, we also evaluate it on the original Synthetic-NeRF [19] dataset where the camera distance remains constant. In this scenario, PyNeRF performs similarly to existing SOTA, as shown in Table 6.

Table 6: **Single-scale results.** We evaluate PyNeRF on single-scale Blender [19]. PyNeRF performs comparably to existing state-of-the-art.

| PSNR | Lego | Mic | Materials | Chair | Hotdog | Ficus | Drums | Ship | Mean |
|---|---|---|---|---|---|---|---|---|---|
| K-Planes [9] | 35.38 | 33.27 | 29.57 | 33.88 | 36.19 | 30.81 | 25.62 | 30.16 | 31.86 |
| TensoRF [6] | 35.14 | 25.70 | **33.69** | **37.03** | 36.04 | 29.77 | 24.64 | 30.12 | 31.52 |
| iNGP [20] | 35.67 | **36.85** | 29.60 | 35.71 | 37.37 | 33.95 | 25.44 | 30.29 | 33.11 |
| Nerfacto [28] | 34.84 | 33.58 | 26.50 | 34.48 | 37.07 | 30.66 | 23.63 | **30.95** | 31.46 |
| PyNeRF | **36.63** | 36.39 | 29.92 | 35.76 | **37.64** | **34.29** | **25.80** | 30.64 | **33.38** |

## B  Additional results

We list results for each individual downsampling level in Table 7 and Table 8 to supplement those shown in Table 1 and Table 2.

Table 7: **Synthetic results.** We average results across Multiscale Blender [2] and Blender-A and list metrics for each downsampling level. All PyNeRF variants outperform their baselines by a wide margin.

| | ↑PSNR | | | | ↑SSIM | | | | ↓LPIPS | | | | |
|---|---|---|---|---|---|---|---|---|---|---|---|---|---|
| | Full Res. | 1/2 Res. | 1/4 Res. | 1/8 Res. | Full Res. | 1/2 Res. | 1/4 Res. | 1/8 Res. | Full Res. | 1/2 Res. | 1/4 Res. | 1/8 Res. | ↓Avg Error |
| Plenoxels [25] | 22.61 | 23.68 | 24.54 | 23.62 | 0.767 | 0.768 | 0.784 | 0.789 | 0.307 | 0.265 | 0.200 | 0.161 | 0.102 |
| K-Planes [9] | 25.14 | 27.03 | 30.26 | 30.75 | 0.807 | 0.840 | 0.896 | 0.925 | 0.225 | 0.163 | 0.085 | 0.053 | 0.046 |
| TensoRF [6] | 25.93 | 28.12 | 31.46 | 30.97 | 0.865 | 0.893 | 0.921 | 0.930 | 0.169 | 0.112 | 0.064 | 0.056 | 0.042 |
| iNGP [20] | 26.90 | 29.14 | 30.89 | 28.49 | 0.865 | 0.905 | 0.947 | 0.947 | 0.152 | 0.095 | 0.047 | 0.054 | 0.032 |
| Nerfacto [28] | 25.35 | 27.26 | 29.78 | 29.09 | 0.809 | 0.840 | 0.893 | 0.917 | 0.214 | 0.158 | 0.094 | 0.068 | 0.049 |
| Mip-NeRF [2] | 32.07 | 33.65 | 34.76 | 35.00 | 0.952 | 0.959 | 0.961 | 0.960 | 0.048 | 0.036 | 0.028 | 0.021 | 0.020 |
| PyNeRF | **33.18** | **35.83** | **37.59** | **38.29** | **0.964** | **0.977** | **0.984** | **0.989** | 0.030 | **0.013** | **0.007** | **0.004** | **0.008** |
| PyNeRF-K-Planes | 33.12 | 35.18 | 36.45 | 36.94 | 0.963 | 0.973 | 0.980 | 0.985 | **0.028** | 0.014 | 0.009 | 0.005 | **0.008** |
| PyNeRF-TensoRF | 32.94 | 35.34 | 36.92 | 37.46 | 0.959 | 0.974 | 0.982 | 0.987 | 0.033 | 0.014 | 0.008 | 0.005 | **0.008** |

Table 8: **Real-world results.** We average results across Boat [24] and Mip-NeRF 360 [3]. As in Table 7, all PyNeRF variants improve significantly upon their baselines.

| | ↑PSNR | | | | ↑SSIM | | | | ↓LPIPS | | | | |
|---|---|---|---|---|---|---|---|---|---|---|---|---|---|
| | Full Res. | 1/2 Res. | 1/4 Res. | 1/8 Res. | Full Res. | 1/2 Res. | 1/4 Res. | 1/8 Res. | Full Res. | 1/2 Res. | 1/4 Res. | 1/8 Res. | ↓Avg Error |
| Plenoxels [25] | 20.69 | 20.70 | 20.98 | 21.93 | 0.627 | 0.543 | 0.547 | 0.640 | 0.661 | 0.607 | 0.525 | 0.364 | 0.128 |
| K-Planes [9] | 20.53 | 20.55 | 20.84 | 21.85 | 0.618 | 0.525 | 0.512 | 0.602 | 0.655 | 0.587 | 0.488 | 0.328 | 0.128 |
| TensoRF [6] | 17.31 | 17.33 | 17.49 | 17.96 | 0.548 | 0.431 | 0.367 | 0.384 | 0.748 | 0.714 | 0.662 | 0.552 | 0.190 |
| iNGP [20] | 19.53 | 19.83 | 16.06 | 20.86 | 0.598 | 0.504 | 0.489 | 0.574 | 0.670 | 0.610 | 0.517 | 0.351 | 0.146 |
| Nerfacto [28] | 21.37 | 21.42 | 21.81 | 23.15 | 0.629 | 0.558 | 0.575 | 0.688 | 0.594 | 0.512 | 0.389 | 0.226 | 0.110 |
| Mip-NeRF 360 w/ GLO [3] | 21.73 | 21.72 | 22.13 | 23.65 | **0.650** | **0.597** | **0.628** | **0.736** | **0.518** | **0.427** | **0.309** | **0.165** | 0.100 |
| Mip-NeRF 360 w/o GLO [3] | 21.01 | 21.00 | 21.39 | 22.88 | 0.634 | 0.580 | 0.610 | 0.718 | 0.529 | 0.441 | 0.323 | 0.179 | 0.111 |
| Exact-NeRF w/ GLO [14] | 20.72 | 20.73 | 21.04 | 22.34 | 0.637 | 0.571 | 0.583 | 0.674 | 0.559 | 0.478 | 0.378 | 0.237 | 0.121 |
| Exact-NeRF w/o GLO [14] | 20.98 | 20.97 | 21.34 | 22.80 | 0.635 | 0.578 | 0.604 | 0.710 | 0.548 | 0.451 | 0.339 | 0.192 | 0.113 |
| PyNeRF | **22.05** | **22.16** | **22.56** | **23.84** | 0.645 | 0.591 | 0.620 | 0.725 | 0.535 | 0.441 | 0.316 | 0.184 | **0.098** |
| PyNeRF-K-Planes | 21.47 | 21.49 | 21.87 | 23.18 | 0.633 | 0.570 | 0.591 | 0.694 | 0.563 | 0.478 | 0.362 | 0.217 | 0.108 |
| PyNeRF-TensoRF | 20.82 | 20.89 | 21.25 | 22.48 | 0.594 | 0.521 | 0.528 | 0.630 | 0.648 | 0.558 | 0.438 | 0.284 | 0.122 |