# OpenReview forum: "PyNeRF: Pyramidal Neural Radiance Fields"
_NeurIPS.cc/2023/Conference — NeurIPS 2023 poster_

### Official Review · Reviewer_LTbu · 2023-07-05

**Soundness:** 3 good
**Presentation:** 4 excellent
**Contribution:** 3 good
**Rating:** 7
**Confidence:** 4

**Summary:**

This works tackles the problem of antialiasing in grid based Neural Radiance Field representations (e.g. INGP, DirectVoxGo). To this end, a very simple solution is proposed: *instead of training a single NeRF with multiscale features, train separate NeRFs at different resolutions and decide which to use based on the distance of the sample from the camera (size of the pixel footprint)*. Specifically, two levels (closest smaller and larger) are used to predict the color and density, which are then linearly combined into a single prediction per sample along the ray. This requires one additional MLP evaluation per each sample, but is still much more efficient than the multisamples proposed e.g. in ZipNeRF. The proposed method is evaluated on the Multiscale Blender and Multiscale Mip-Nerf360 datasets, where it consistently outperforms the state of the art.

**Strengths:**

In my opinion, the main strength of this paper is its simplicity. The proposed idea is very general and could easily be implemented in the grid based representation at only a minor cost (one additional MLP evaluation per each sample). The increased storage is also not a large problem, as shown in the supplementary, as features can be shared across level resulting only in the overhead of MLP weights (very small compared to the feature volumes). I really like such simple ideas that lead to good performance improvement.

The paper is also well presented and easy to understand, the ablation studies support the main ideas (Pyramid training, simplification of the Laplacian pyramid). I especially liked the analysis in the supplementary material (it is maybe unfortunate to have that only in the supplement and could maybe be moved to the main paper for the camera ready version).

**Weaknesses:**

I only have one minor weakness - sometimes some details are missing, or the description is slightly ambiguous. For example, the number of levels, size of the features per level, MLP dimensions are missing. In the supplementary, it is not fully clear to me what the shared INGP table denotes? In the original INGP each level has its own hash-table, does this imply that a single one is used for all level (what is the dimension and number of the features in this hash-table?)

**Questions:**

**Comments - potential improvement**:
- The proposed method is very general, but it is only evaluated on the INGP representation. While TensorRF is used in the ablation in the supplementary, it is not clear if the proposed multilevel training also improves its performance. It would be great to support the claims of the generality by applying the ideas to at least one more representation.

**Questions**:
- The gradients for each camera sample are only propagated back to 2 levels. If the camera poses are very different, this could mean that in some areas the finest levels are never supervised - what happens if the novel view is closer than camera of the training view? It would mean that the levels that are sampled were never supervised or?

- In the representation in the main paper, I don't fully understand why the storage requirements are larger? Is it because there is no concatenation of the features, and hence the feature vectors of each level are of higher dimension? It would be good to clarify this

- If I understand correctly, the method proposed in the supplementary with the shared features across the levels is even simpler (only adds L-1 MLPs to the formulation) and achieves comparable performance. Is there are good reason to not make that the main method? I guess it is not trivial to share the features in the methods that store them explicitly (tensorRF or DirectVoxGo).

**Limitations:**

The authors describe the limitations of their method and potential negative societal impacts adequately.

---

> ### Author Rebuttal · Authors · 2023-08-08
>
> Thank you for reviewing our paper and for the helpful comments!
>
> **Generalization.** With regards to improvements being marginal for other backbones such as TensoRF, we admittedly focused on hashtable approaches for our original submission. As the authors of TensoRF state, it is mainly designed for bounded scenes [1] and our initial evaluation of TensoRF on unbounded 360 views yielded poor results. In response to reviewer concerns, we have added optimizations not present in the original TensoRF method (such as Mip-NeRF 360's scene contraction [2]), improving the baseline TensoRF from 14.75 to 17.21 dB PSNR, and now see a large improvement when combined with PyNeRF (21.35 dB). We also ran experiments with K-Planes which show similar improvements with PyNeRF (2–6 dB gain in PSNR). We have included updated tables in our rebuttal PDF.
>
> **Unsupervised levels.** You are correct that one of the limitations of PyNeRF is that performance will degrade when zooming in and out of areas that have not been seen at training time. "Zoom-out" (rendering far-away views) can be handled by simulating far-away views during training, by simply adding downsampled training images. "Zoom-in" (rendering nearby views) is fundamentally challenging since one cannot easily simulate such views without hallucination-based methods such as super-resolution. Instead, we can force renderings to query only the finest level that was supervised at train-time (by maintaining an occupancy grid-like data structure), which should result in the same blurry artifacts that typical NeRF approaches exhibit during excessive "zoom-in". We will note these limitations in the camera-ready paper.
>
> **Storage.** To clarify why storage requirements are larger with our base method, each level is backed by an entirely separate feature grid (which in the case of iNGP happens to be a multi-resolution grid). In the general case (which doesn't assume anything about the underlying storage of the NeRF backbone), we create a separate table for each level (up to the target resolution of the level). In the case of iNGP, this is redundant and we can just use a single multi-resolution table for all levels. As you correctly assumed, we don't do this in the general case since this does not apply to backbones such as TensoRF or Plenoxels. That said, the use of a multi-resolution backbone is particularly natural for our method, and so we plan to present both the general and storage-optimized variant (for multi-resolution backbones) in the main paper.

---

> > ### Comment · Reviewer_LTbu · 2023-08-13
> > **Response to the rebuttal**
> >
> > Thank you for clarify some details, I think that adding the discussion on the zoom in/out effect to the supplementary would help strenghten the paper.
> >
> > **Storage**:
> >
> > This is still somewhat confusing to me, but let me see if I understand correctly. Let's assume an INGP configuration with 16 levels, the original works will store 16 distinct hash tables (one per each level) with e.g. $2^{19}$ features each of dimension 2. When querying the features, features of each level will be interpolated trilenearly first and then concatenated to a 32 dim feature vector (16 levels of dim 2).
> >
> > If I understand correctly in your base case you have the following hash tables: for grid resolution 1 (1 hash tables with dim xx), for grid resolution 2 (2 hash tables with dim xx), ... for grid resolution 16 (16 hash tables with feature dim xx)? This would also mean that the dimension of the input to the MLP changes based on the level? What is the xx in this case?
> >
> > In the storage optimized version, you instead have the same as INGP, i.e. 16 distinct hash tables (one per each level) with e.g. $2^{19}$ features each of dimension 2. And you only index into a different MLP based on the grid resolution?

---

> > > ### Author Response · Authors · 2023-08-14
> > >
> > > Thanks for your response!
> > >
> > > Your understanding of storage for the base case is largely correct, but with two minor modifications. First, because very few ray samples map to the coarsest grid solutions, we chose to build separate grid resolutions for only the finest levels 8 - 16 rather than all levels 1 - 16. In practice, when sampling a 3D point along a ray that should map to a coarse level 1-7, we naively query level 8. We train separate MLPs per grid resolution - the first MLP would get 2\*8=16 features as input, the second would get 2\*9=18 features as input, etc. Second, we explored a variant where each of the 8 PyNeRF grid resolutions is backed by an internal 16-level multiresolution hash table with smaller scale factors. Here, the input feature size to each MLP is always fixed to 2*16=32. This variant performs slightly better at the cost of more storage (and is the base case reported in the paper). But importantly, the difference is minor and the storage-optimized version provides the best performance-storage tradeoff.
> > >
> > > Your understanding of the storage-optimized version is correct!
> > >
> > > Thank you once more for the questions - adding these clarifications will hopefully make for a stronger paper!

---

> > > > ### Comment · Reviewer_LTbu · 2023-08-15
> > > >
> > > > Thank you for clarifying the implementation details. In my opinion, it would be very important to include these in the revised version of the paper.

---

### Official Review · Reviewer_pHej · 2023-07-06

**Soundness:** 2 fair
**Presentation:** 3 good
**Contribution:** 2 fair
**Rating:** 5
**Confidence:** 5

**Summary:**

PyNeRF replaces the implicit representation in Mip-NeRF with a voxel-based representation method, which combines the cone sampling method and explicit structural representation by interpolating on different voxels based on coordinates of different scales. This method can be easily applied to existing accelerated NeRF methods. PyNeRF improves training speed by 60 times compared to Mip-NeRF while ensuring anti-aliasing effects and reducing errors by 20%.

**Strengths:**

* Experiments are performed on multiple datasets, including synthetic, real, and large-scale datasets, which demonstrate the superior performance and training speed of the proposed method. PyNeRF combines the anti-aliasing capability of MiP-NeRF with the fast fitting of explicit structure, showcasing the advantages of the two methods.
* Three strategies are adopted for the interpolation method (GaussPyNeRF, LaplacianPyNeRF, pyNeRF), demonstrating that the PyNeRF approach can save storage, improve rendering speed, and ensure the same rendering quality as interpolating on multiple voxels. This verifies the rationality of PyNeRF's interpolation strategy.

**Weaknesses:**

* The paper states that the proposed method can be applied to existing accelerated NeRF approaches (L9). However, the results show only slight improvements when applied to other acceleration methods (PyNeRF - TensoRF-CP, PyNeRF - TensoRF-VM in Table 5 of the supplementary). It appears that the use of hash tables is the only approach that shows significant improvement.
* Although the multi-scale sampling and interpolation methods in this paper play a role in anti-aliasing and accelerating training speed, they also increase the consumption of storage space. According to Table 5, using a shared hash table can achieve the same performance, so why not use a shared hash table? This would save storage space.
* Moreover, adopted methods, e.g. multiscale sampling and multi-resolution voxels, have been proposed and widely used in many existing methods, which makes the novelty somewhat limited.

**Questions:**

* Further analysis is needed to demonstrate why the gain on TensoRF is marginal.
* The feasibility of the method can be tested on more accelerated NeRF approaches. The storage size of this method may be the main factor that impedes its wider application.

**Limitations:**

The authors have addressed the limitations.

---

> ### Author Rebuttal · Authors · 2023-08-08
>
> Thank you for reading our paper and for the constructive feedback.
>
> **Contribution.** To the best of our knowledge, our work is among the first to combine fast NeRF rendering with anti-aliasing. We agree that testing on more accelerated NeRF approaches would improve our paper. To that effect, we present additional results with K-Planes in the global rebuttal PDF which show a 2–6 dB improvement over the standard K-Planes baseline. Per your suggestion, we also reexamined our TensoRF results. As its authors state, it is designed for bounded scenes [1] and our initial results on unbounded scenes were especially poor. By adding optimizations not present in the original TensoRF method (namely Mip-NeRF 360's scene contraction [2]), we're able to improve baseline TensoRF performance (from 14.75 to 17.21 dB PSNR on unbounded scenes) and obtain a large improvement when combined with PyNeRF (21.35 dB). We will add these updated numbers and experiment details to the camera-ready version.
>
> **Storage.** As noted in the Limitations section, we agree that storage space is a potential tradeoff of our approach. Our intent with the shared hashtable approach in the supplement was to provide a mitigation for methods that use multi-resolution structures (iNGP, K-Planes), at the cost of generalizability to those that don't (TensoRF, Plenoxels). We will make this more clear in the revised paper by presenting both the general PyNeRF method and the storage-optimized version (for multi-resolution backbones) in the main paper.
>
> References:
>
> [1] Anpei Chen, Zexiang Xu, Andreas Geiger, Jingyi Yu, and Hao Su. 2022. TensoRF: Tensorial Radiance Fields. In ECCV 2022.
>
> [2] J. T. Barron, B. Mildenhall, D. Verbin, P. P. Srinivasan, and P. Hedman. Mip-NeRF 360: Unbounded anti-aliased neural radiance fields. In CVPR, 2022.

---

> > ### Comment · Reviewer_pHej · 2023-08-18
> > **Thanks and Final Rating**
> >
> > Thanks for the authors' rebuttal which has addressed my concerns. I would like to raise my rating. Besides, it is highly recommended to include the additional results and contents in the final version.

---

### Official Review · Reviewer_JoFm · 2023-07-06

**Soundness:** 3 good
**Presentation:** 3 good
**Contribution:** 3 good
**Rating:** 6
**Confidence:** 4

**Summary:**

The authors introduce a pyramidal radiance field reconstruction method, which reuses multi-scale feature grid representation and area matching algorithm for level indexing. Specifically, the method trains a pyramid of models at different scales and interpolates point features between neighboring levels determined by the cone size to which the point belongs to. The effectiveness of the proposed method is evaluated on the multi-resolution version of the blender, ADOP and Argoverse sensor dataset. The experiments show that the proposed method is able to provide better rendering quality while preserving high reconstruction speed.

**Strengths:**

1. The motivation is clear. The manuscript addresses an interesting and important anti-aliasing problem for grid-based representation. It's good to a paper on that.

2. The manuscript is well-written. Figure 1 and algorithm 1 are very helpful for understanding the manuscript, good job.

3. The quantitive results reported in the tables are quite good.

**Weaknesses:**

1. The quantitive result on the standard resolution is missing, such as single resolution for nerf synthetic, mipnerf 360, leading it hard to judge the general performance.

2. The optimization progress is unclear to me. Is the different level feature optimized with the specific scale of the dataset (L113-115) or supervised only on the final rendering (L129)?

3. Experiment details are missing, such as level numbers used for each dataset, the grid resolutions, and the feature channel, representation (dense grid?).

4. Since the manuscript is focused on anti-aliasing, but I could not find the zoom-in results in the appendix video, are there flickers when rendering sequences?

**Questions:**

1. Can this representation be trained and evaluated on a standard multi-view dataset?

2. Table 1 and 2, any insights on why the quantitive (Plenoxels, K-Planes, TensoRF and iNGP)) and the rendering video of iNGP is much worse than the result reported in the original papers? Is that because they are evaluated on multi-resolution? In this case, I would expect the rendering quality to be the same.

**Limitations:**

Yes, the authors provide a limitation section and make sense to me.

---

> ### Author Rebuttal · Authors · 2023-08-07
>
> Thank you for reading our work and for the constructive feedback!
>
> **Single vs multi-resolution.** You ask why existing methods such as iNGP perform worse on multi-resolution datasets. Mip-NeRF [1] originally points out that prior work struggles on scenes where the same scene content is viewed from different distances across training and test cameras. For scenes when the camera distance remains roughly constant, PyNeRF performs similarly to existing SOTA, as shown in Table 3 in our rebuttal PDF. mip-NeRF [1] simulates views from different distances by adding 2×–8× downsampled images to training and test sets. Similar to mip-NeRF [1], our rebuttal PDF includes performance numbers on each individual downsampling level in Tables 1 and 2.
>
> **Optimization.** You ask whether we train different pyramid levels separately or jointly with the final rendering loss (L129). We train jointly; different samples along a camera ray typically map to different levels of the hierarchy (Figure 3(a)), but all samples are composited into a single color that is supervised with a per-pixel L2 loss.
>
> **Experiment details.** We will add more details to our camera-ready version. Our main experiments use a PyNeRF hierarchy of 8 levels corresponding to resolutions 128 to 16,384. Each level of the hierarchy is backed by a feature hashtable (same as iNGP), followed by small density and color MLPs (1 layer/64 channels and 2 layers/128 channels, respectively). We will commit to releasing code/data upon acceptance to aid reproducibility. We will also add videos with zoom in effects (which do not exhibit flickering).
>
> References:
>
> [1] J. T. Barron, B. Mildenhall, M. Tancik, P. Hedman, R. Martin-Brualla, and P. P. Srinivasan. Mip-NeRF: A multiscale representation for anti-aliasing neural radiance fields. In ICCV, 2021

---

> > ### Comment · Reviewer_JoFm · 2023-08-15
> > **final rating**
> >
> > Thanks for the rebuttal and clarification, I am leaning toward the original "Weak Accept" rating.

---

### Official Review · Reviewer_PaSq · 2023-07-08

**Soundness:** 3 good
**Presentation:** 3 good
**Contribution:** 3 good
**Rating:** 7
**Confidence:** 5

**Summary:**

This work presents a method for anti-aliased renderings for grid-based NeRF representations by jointly optimizing a hierarchy of coarse-to-fine grids. The idea is neat and well justified by empirical evaluations that show quantitative and qualitative improvements over baselines.

**Strengths:**

1. The proposed method seems simple and easy to reproduce.

2. Experiments are extensive, and demonstrate the effectiveness of the proposed method.

**Weaknesses:**

1. The method is built upon the nerfstudio library. But the empirical results of that base model seems missing. Would be great to include those results to help better see the improvements made by the proposed anti-aliasing techniques.

**Questions:**

1. It says that the method is implemented on top of the nerfstudio library. Which specific model was built upon? Is it the nerfacto one, or others?

2. What do the numbers 11.2,8.6,5.7 mean in Fig. 3(a)?

3. Is the MLP shared across the different mipmap levels? What is the MLP size?

4. Does the method apply to other grid-based representations like TensoRF, or K-planes?


**Limitations:**

Limitations are adequately addressed.

---

> ### Author Rebuttal · Authors · 2023-08-08
>
> Thank you for reviewing our paper! We’re glad that you appreciate the simplicity and effectiveness of our approach.
>
> **Nerfstudio.** Our implementation is indeed built on the library (we will release our code as a plugin) and is closest to Nerfacto. We list comparison numbers in the tables attached as part of our global response which we will add to our camera ready. We show a 1-7db improvement in PSNR.
>
> **MLPs.** We use separate MLPs per level. We use a 64 channel density MLP with 1 hidden layer followed by a 128 channel color MLP with 2 hidden layers. We use the same size MLPs across the iNGP/K-Planes/Nerfacto baselines for fairness.
>
> **Other backbones.** A strength of our approach is its applicability to other grid-based representations - we list results with a K-Planes backbone as part of our global rebuttal which shows a significant improvement over its baseline (2-6 db PSNR) and report a 4-6 db improvement in PSNR when using TensoRF.
>
> **Fig 3(a).** The numbers are to illustrate the different hierarchy levels each ray sample could fall under (Equation 1 in the paper) - we will clarify.

---

### Official Review · Reviewer_LPiw · 2023-07-09

**Soundness:** 3 good
**Presentation:** 3 good
**Contribution:** 3 good
**Rating:** 6
**Confidence:** 4

**Summary:**

This paper presents a method to address the aliasing artifacts in grid-based NeRF. It introduces a pyramid of grids of different resolutions to represent a scene. To query the color and density of a 3D point with a certain integration volume, the method finds the two pyramid levels that best describe the point and performs a weighted combination between the outputs of the two levels. Experiments on both synthetic and real datasets demonstrate the advantages of PyNeRF over previous methods in rendering quality and speed.

**Strengths:**

- How to address alias artifacts is an important problem in NeRF. While mip-NeRF proposes an elegant solution for positional-encoding-based NeRF, extending this to grid-based NeRF is not straightforward. This paper presents a reasonable and novel approach to address the aliasing issue in grid-based NeRF. This is achieved by maintaining a grid pyramid with different levels of resolutions and only using suitable levels during rendering.

- The proposed PyNeRF shows clear advantages over previous approaches, as demonstrated in experiments. The rendering accuracy is significantly better than other grid-based NeRF and in most cases better than mip-NeRF. Qualitative results also show that PyNeRF better captures fine details. The running speed is on par with other grid-based NeRF and is much faster than MLP-based NeRF.

- The effectiveness of the proposed design choice is verified in ablation studies.



**Weaknesses:**

- The technical contribution is weak, as the proposed method is rather simple and does not involve much technical challenge. The iNGP already has a multi-resolution grid and this work simply chooses to use the two suitable for the scale of the point. Although this strategy is shown to be effective, I doubt if the technical contribution is enough to be presented at NeurIPS.

- Some descriptions are not clear or reasonable. For example:

  - The sentence on Line 121 and 122 is unclear. It's not well explained how P(x) is calculated. It seems to be the projected area from the voxel to the 2D plane. If this is the case, then it is a somewhat odd design. A voxel is a cube, thus its projected area to the 2D plane is view-dependent. But the selection of scale level should not depend on view but depend on the distance between the 3D point and the camera.

  - The first version of the method is called GaussPyNeRF. However, it is not explained how the method is related to Gaussian. I do not see a clear connection.

  - Line 156: "We can quickly train these along with our initial low-resolution model and then use them to train higher-resolution levels in a sample-efficient manner." It's not clearly described how this is done.



**Questions:**

- Authors can respond to my concerns mentioned in weaknesses.

- Line 155: "many contemporary NeRF methods use occupancy grids or proposal networks to generate refined samples near surfaces". I suggest adding related citations here.

- Line 35: fast-rendering approach -> grid-based approach. Fast rendering includes not only grid-based approaches but also others like point-based approaches. This paper targets grid-based approaches.

- The title does not mention anti-aliasing, which is the main problem to be solved in this paper. A minor suggestion is to mention it in the title.

**Limitations:**

Yes.

---

> ### Author Rebuttal · Authors · 2023-08-08
>
> Thank you for reviewing our paper and for the valuable feedback!
>
> **Contribution.** Rather than seeing the simplicity of our method as a weakness, we humbly agree with Reviewer LTbu's statement that "the main strength of this paper *is* its simplicity". As you state, our simple method significantly outperforms the state of the art.
>
> **Lines 121-122.** We calculate P(x) solely based on the distance between the 3D point and camera similar to Nerfstudio's utility method [1], and then compare it to the resolution hierarchy described in lines 115-116. For example, for a point along the camera ray with a projected pixel area of 0.125, and a PyNeRF hierarchy corresponding to resolutions [2, 4, 8, 16], we query the third level of the hierarchy (with resolution 1 / 8 = 0.125). We will clarify in the camera-ready.
>
> **GaussPyNeRF.** Our intent was to draw an analogy between the resolution and details captured by the different levels in our NeRF hierarchy (Fig. 2) and those characteristic of Gaussian pyramids used in classic image processing [2]. We will clarify.
>
> **Line 155.** SOTA methods either use a proposal network (Mip-NeRF 360, Nerfacto, SUDS, K-Planes) or an occupancy grid (iNGP) to learn a coarse 3D representation and guide sampling near surfaces (vs wastefully querying empty space). Occupancy grids / proposal networks are usually trained jointly with the main NeRF network, which takes significant time at city scale. Selectively training in a coarse-to-fine manner with the low-resolution PyNeRF levels as described in Sec. 4.4 speeds up convergence.
>
> **Line 35.** Agreed - we will amend!
>
> References:
>
> [1] https://github.com/nerfstudio-project/nerfstudio/blob/9b6010ea0d20e7ca2a68496bb33d6dcd03bf9b91/nerfstudio/cameras/cameras.py#L849
>
> [2] E. Adelson, C. Anderson, J. Bergen, P. Burt, and J. Ogden. Pyramid methods in image processing. RCA250 Eng., 29, 11 1983.

---

> > ### Comment · Reviewer_LPiw · 2023-08-15
> > **Response to the rebuttal**
> >
> > Thank you for the clarifications. The rebuttal addresses most of my concerns and I remain positive about this paper. Please make these revisions accordingly in the revised version.

---

### Author Rebuttal · Authors · 2023-08-08

We are glad that reviewers agree that we address "an interesting and important anti-aliasing problem," (JoFm), appreciate "simple ideas that lead to good performance improvement," (LTbu), and acknowledge that "experiments are extensive, and demonstrate the effectiveness of the proposed method." (PaSq)

Two common concerns reviewers raised were that our contribution would be stronger with additional results on non-iNGP backbones (PaSq, pHej, LTbu) and that we should include additional details on our model architecture and experimental setup (LTbu, JoFm).  We present additional experiments with K-Planes and TensoRF backbones in the attached PDF (Tables 1 and 2). Both variants show clear improvements over their base methods. We will gladly add more model and hyperparameter details to the camera ready and commit to releasing code and data to address any remaining reproducibility concerns.

---

### Decision · Program_Chairs · 2023-09-21

**Decision:**

Accept (poster)

**Comment:**

The paper presents a method for multiscale neural radiance fields that provides improved rendering via approximate anti-aliasing. Specifically, multiple feature grids with different resolutions are used to render from the radiance field, and the coarser resolution grids are used to render regions with lower spatial resolutions. All reviewers were positive towards accepting the paper, though some reviewers requested additional experimental details and evaluating the method on different NeRF backbones (e.g., TensoRF and K-Planes in addition to Instant NGP). The authors addressed these concerns in the rebuttal, and the AC finds that the paper is suitable for acceptance. The authors should include the results from the rebuttal in the camera ready version.